# Acceptability of Patient Portals and Phone Consultations in Hybrid Primary Care: A Slovenian Multi-Centre Pilot Study

**DOI:** 10.3390/healthcare13212662

**Published:** 2025-10-22

**Authors:** Matic Mihevc, Snežana Đurić, Marija Petek Šter

**Affiliations:** 1Department of Family Medicine, Medical Faculty, University of Ljubljana, 1000 Ljubljana, Slovenia; 2Primary Healthcare Research and Development Institute, Community Health Center Ljubljana, 1000 Ljubljana, Slovenia; 3Community Health Center Tržič, 4290 Tržič, Slovenia

**Keywords:** digital tools, patient portal, teleconsultation, acceptability, digital literacy, quality of life, hybrid care

## Abstract

**Background/Objectives**: With digital transformation, patient portals and phone consultations are increasingly integrated into hybrid primary care workflows that combine in-person and remote services. This study aimed to assess the acceptability of these tools among patients and identify factors associated with acceptability. **Methods**: Between April and June 2025, a multicenter cross-sectional survey was conducted in four primary healthcare centers in Slovenia. The sample included 214 people who had used both patient portals and phone consultations within the previous 12 months. Data collected covered socio-demographic and clinical profile, digital communication skills, quality of life, and annual use of digital tools. Acceptability was assessed using the Theoretical Framework of Acceptability (TFA) tool. Univariate and multivariable linear regression analyses were performed to identify factors associated with acceptability. **Results**: Among the 214 participants (mean age 42.9 ± 14.1 years; 61.2% female), both patient portals and phone consultations were generally acceptable, with similar overall TFA scores (3.9/5). Patient portals were considered as significantly less time-consuming and better for communication, whereas phone consultations were preferred for accessibility and reliability. Multivariable analyses showed that higher digital communication skills and better quality of life predicted greater acceptability for both methods, whereas lower education level and more frequent use were associated with higher acceptability of phone consultations. **Conclusions**: Acceptability of patient portals and phone consultations varies by education, digital communication skills, and quality of life. This highlights the need for personalized hybrid care solutions. Healthcare providers should offer flexible digital options, invest in digital literacy programs, and develop interoperable eHealth infrastructure to enable safe and sustainable integration of advanced tools such as video consultations.

## 1. Introduction

Healthcare delivery is undergoing a profound transformation as digital technologies redefine how patients and providers interact. Central to this shift are hybrid primary care models, which combine traditional in-person visits with remote digital services such as patient portals and phone consultations. These models offer greater flexibility, improving efficiency, accessibility, and responsiveness to diverse patient needs [1,2,3].

Traditionally, primary care relied heavily on face-to-face consultations, which became increasingly challenging due to rising patient demand and limited healthcare resources. The COVID-19 pandemic accelerated the adoption of digital health solutions, demonstrating how telemedicine can maintain care continuity while improving access and quality [2,3,4,5,6].

In Slovenia, primary healthcare centers (PHCs) have followed global trends by implementing digital tools to improve access and streamline service delivery. Many PHCs have set up online portals where patients can book appointments, request e-prescriptions, get medical certificates, manage sick leave, and handle other administrative tasks. This reduces provider’s workload and speeds up administrative processes [7,8,9,10].

Beyond administrative functions, remote consultations have become a core component of hybrid care, facilitating timely medical advice, follow-ups, and chronic disease management without requiring physical visits. Phone consultations, while not strictly digital tools, are widely used as a remote communication modality because they are simple, accessible, and do not require advanced digital skills. They allow patients who may have limited digital literacy, poor access to digital devices, or difficulties navigating online platforms to still receive care remotely. In contrast, patient portals engage users with more advanced digital communication skills, offering interactive features and direct access to personal health information [3,11,12,13].

As digital tools become increasingly embedded in primary care, understanding patients’ experiences and perceptions is essential. A central concept in this regard is acceptability, which reflects how appropriate and suitable healthcare intervention is from the patient’s perspective, considering their needs, expectations, and experiences [14,15]. This concept includes both cognitive and emotional responses, covering factors such as satisfaction with the service, perceived time burden, effectiveness, ethical alignment, clarity of purpose, ease of use, and any opportunity costs involved [4,16].

However, previous studies have often focused on technical feasibility or usage rates, with less attention given to the multidimensional nature of acceptability from the patient’s viewpoint. Many have overlooked how individual factors such as digital literacy, health status, and socio-demographic characteristics influence patient acceptance of hybrid care models. Addressing these gaps is critical for designing digital health interventions that are truly patient-centered and equitable [4,15,17,18].

The aim of this study is to assess the acceptability of patient portals and phone consultations as complementary forms of patient care in hybrid primary care workflows in Slovenia and to identify patient-related factors associated with lower acceptability.

## 2. Materials and Methods

### 2.1. Study Design

We conducted a pilot multi-center cross-sectional survey among adult visitors to family medicine practices in Slovenia that was approved by the Medical Ethics Committee of Slovenia and compliant with the Declaration of Helsinki.

### 2.2. Study Setting

The study took place in four PHCs in western Slovenia region, chosen for their varied demographic characteristics. PHC Kranj (57,000 residents) and PHC Jesenice (21,000) represent urban areas, while PHC Bled (8000) and PHC Tržič (15,000) reflect rural communities. Together, these centers serve a population of approximately 100,000 people. This region was an early adopter of a patient portal in 2022. Following inclusion, healthcare workers underwent training delivered by the patient portal providers and subsequently offered individualized instruction to patients on portal use. Additionally, community workshops and online support resources were provided to enhance patient education. Since then, patients and healthcare providers have regularly used the portal, allowing observation of long-term adoption and stable user experience.

### 2.3. Study Population and Patient Recruitment

We recruited adult patients aged 18 years or over who attended family medicine practices between April and June 2025. The inclusion criteria were: (a) use of both patient portals and phone consultations at least once within the previous 12 months, and (b) ability to understand and respond to survey questions in Slovene. The exclusion criteria were: (a) cognitive impairment, severe psychiatric illness, or medical conditions that prevented effective use of either service, (b) significant sensory impairment (e.g., vision or hearing impairment) that limited the use of patient portals or phone consultations, and (c) no prior experience with patient portals or phone consultations within the last 12 months.

Using convenience sampling, we enrolled consecutive eligible patients until the target of 250 was reached. All participants received study information and provided written informed consent upon inclusion.

### 2.4. Healthcare Context and Hybrid Primary Care Model in Slovenia

Primary care in Slovenia operates within a complex and partially digitalized healthcare system. The country has invested in digital health infrastructure, such as the national eHealth portal, which provides secure access to prescriptions, medical reports, and referrals. However, the portal primarily functions as an information platform, with limited capacity for direct service delivery, such as scheduling appointments or requesting prescriptions, except for certain specialist consultations. This limitation reflects broader challenges in Slovenian primary care, including fragmentation between administrative, clinical, and communication functions [3,10,19].

To address these gaps, hybrid care models have emerged, combining digital tools with traditional care channels to improve accessibility, efficiency, and continuity of care. Two key patient portals, doZdravnika and Gospodar Zdravja, evaluated in this study, serve as complementary tools for family medicine teams and patients. These portals provide secure interfaces for appointment scheduling, prescription renewals, sick leave management, referral requests, and direct messaging between patients and clinicians, effectively bridging communication and administrative gaps left by the national system. Alongside these digital tools, phone consultations remain an important component of hybrid care. Formally reimbursed during the COVID-19 pandemic, phone consultations allow real-time discussion of symptoms, treatment planning, test result interpretation, and follow-up care, without requiring in-person visits. Video consultations, while technologically feasible, remain underutilized due to a lack of national guidelines and full integration into primary care workflows [10,19].

The Slovenian hybrid primary care workflow, as illustrated in Figure 1, reflects a multi-layered approach. Patients initiate actions such as administrative requests, symptom reporting, information seeking, test orders, appointment scheduling, or home monitoring. These actions are first assessed via phone triage or online portals and then directed into care delivery pathways ranging from in-person visits and phone consultations to telemonitoring. Color-coded pathways in the workflow emphasize the diversity of patient needs and the multiple routes through which hybrid care can be delivered. Video consultations and telemonitoring are still in the pilot phase, reflecting ongoing efforts to fully integrate these modalities into routine practice [4].

### 2.5. Data Collection

Data were collected using a structured paper-based questionnaire, which included socio-demographic and clinical information, assessment of digital communication skills, health-related quality of life (HRQoL) and digital tool usage, as well as measures of acceptability. All participants completed the survey during their clinic visit, and data were subsequently verified in medical records where applicable.

#### 2.5.1. Socio-Demographic Profile and Clinical Data

Participants reported age, gender, education, marital status, and region. Clinical data on chronic conditions, medications, and medical visits in the past year were obtained and verified in medical records.

#### 2.5.2. Digital Communications Skills and Digital Tools Use

Digital communication skills were self-evaluated using the question: “How well do you manage using a computer, telephone, or tablet for everyday tasks such as sending e-mails, searching for information, or using apps?” Responses were recorded on a 5-point Likert scale. The question was developed with reference to the European Digital Competence Framework [20], focusing on communication domain relevant to the use of a patient portal and phone communication. The question was pilot tested with five patients to ensure clarity, and its content was reviewed for validity by two experts in digital health. In our study, digital communication skills were considered low-moderate if rated 3 or below.

#### 2.5.3. Health-Related Quality of Life

HRQoL was measured using the EQ-5D-5L questionnaire, which includes the descriptive system and the Visual Analogue Scale (EQ VAS) [21]. The descriptive system assesses five health dimensions, namely mobility, self-care, usual activities, pain or discomfort, and anxiety or depression on a five-level scale from no problems to extreme problems. Scores were converted to utility values using a Slovenian value set [22], ranging from −1.09 (worst) to 1.00 (best). Internal consistency was acceptable with a Cronbach’s α of 0.78. The EQ VAS is a 0 to 100 scale where participants rate their overall health, with 0 as the worst and 100 as the best perceived health. In our study, low HRQoL was classified as being in the bottom 25% of the obtained scores.

#### 2.5.4. Acceptability Tool

To assess acceptability of online portal and phone consultations, we created a tool based on the Theoretical Framework of Acceptability (TFA) [14]. TFA defines acceptability using seven domains: affective attitude, burden, perceived effectiveness, ethicality, coherence, self-efficacy, and opportunity costs (Figure 2). Each domain included one to three statements rated on a five-point Likert scale from one (strongly disagree) to five (strongly agree). Statements were developed from the TFA, previous studies on acceptability of digital tools [4,16,23], and the local context [19]. The tool was pilot tested with five patients to ensure clarity, and its content was reviewed for validity by three experts in primary care. It demonstrated high internal consistency, with a Cronbach’s α of 0.94 for both patient portals and phone consultations. In our study, low acceptability was classified as being in the bottom 25% of the obtained scores.

### 2.6. Sample Size Calculation

The study aims to estimate the mean acceptability score of patient portals/phone consultations on a 5-point scale in a population of approximately 100,000 users. Previous studies estimate the standard deviation of scores at 0.4 [4,16], indicating low variability. Using this standard deviation, a 95% confidence level, and a margin of error of ± 0.05, the required sample size was calculated with finite population correction: $\frac{N\;x\;z^{2}x\;\delta^{2}}{e^{2}\;x\;\left(N-1\right)+\;z^{2}x\;\delta^{2}}$, where N = 100,000, z = 1.96, δ = 0.4, and e = 0.05. This results in a sample size of about 245 participants.

### 2.7. Statistical Analysis

Data were analyzed using IBM SPSS Statistics Version 25.0 (IBM Corp., Armonk, NY, USA). The variable distribution was assessed with Q–Q plots and normality tested by the Shapiro–Wilk test. Numerical data were summarized as means and standard deviations if normally distributed, or medians with interquartile ranges if not. Between-group differences were tested using independent-samples t-tests for normal data and Mann–Whitney U tests for non-normal data. Within-group comparisons used paired-samples t-tests. Categorical variables were compared with chi-square tests. Multivariate linear regression was conducted to examine associations with acceptability of phone consultations and patient portals. Predictor variables were selected based on univariate significance, prior studies [4,11,12,17], and expert judgement. Models were adjusted for gender, age, and educational level. Multicollinearity was assessed using variance inflation factors, with all values below 1.8 indicating no concern. The results are presented as regression coefficients, 95% confidence intervals, and *p*-values, with significance at *p* < 0.05.

## 3. Results

### 3.1. Socio-Demographic and Clinical Profile

We initially enrolled 259 patients in the study. After excluding incomplete questionnaires that made it impossible to assess the primary outcome, 214 participants were included in the final analysis. There were no significant differences in socio-demographic and clinical characteristics between those who were included and those who were excluded (Appendix A).

Among the 214 participants, the majority were female (61.2%), with a nearly even distribution between urban (48.6%) and rural (51.4%) residents (Table 1). The largest age group was 35–44 years (30.8%), and more than half (51.9%) had completed secondary school. Self-assessed digital communication skills were generally high, with a median score of 5/5. The most reported health conditions were hypertension (8.9%), diabetes (6.5%), and depression (5.6%). Quality of life was rated highly, with a median EQ-5D-5L score of 0.96 and an EQ-VAS score of 80.

Higher acceptability for patient portals and phone consultations was associated with having hypertension, higher digital communication skills, taking more daily medications, better quality of life scores, and among males. No significant differences were observed based on age; living area; or the presence of diabetes, depression, or anxiety.

### 3.2. Acceptability Rates

Overall acceptability was moderate for both patient portals and phone consultations, with a mean TFA score of 3.9/5 and no significant difference in total scores (Table 2). However, patient portals were seen as less time-consuming, better at supporting smooth communication, and increasing patients’ confidence in communicating via the portal. In contrast, phone consultations were rated as more accessible for users with varying digital skills and more reliable in avoiding technical issues. No significant differences were found in satisfaction, perceived effectiveness, privacy, or opportunity costs.

The radar plots (Figure 3) show that higher digital communication skills and higher quality of life were linked to consistently higher ratings for patient portals across most acceptability domains, with many differences statistically significant, while phone consultations scores were largely unaffected. Education showed a different pattern: in patient portals, participants with primary or secondary education often rated domains such as coherence and self-efficacy lower, whereas in phone consultations, higher education was associated with lower ratings in several domains.

### 3.3. Determinants of Acceptability

Multivariable linear regression models identified determinants of acceptability for patient portals (Figure 4) and phone consultations (Figure 5). For portals, higher digital communication skills (β = 0.165, *p* = 0.001) and quality of life (β = 0.904, *p* = 0.007) were significant determinants of higher acceptability, with no effects from gender, age, education, rural residence, number of pills/day or presence of hypertension. For phone consultations, higher digital communication skills (β = 0.090, *p* = 0.049), quality of life (β = 0.666, *p* = 0.038), frequency of use (β = 0.008, *p* = 0.037), and lower education (β = 0.223, *p* = 0.023) predicted higher acceptability, with no effects from other variables.

## 4. Discussion

### 4.1. Principal Findings and Comparison with the Existing Literature

The study found that both patient portals and phone consultations were generally acceptable to patients, with similar overall scores (3.9/5). Portals were seen as less time-consuming and more effective for communication, while phone consultations were preferred for their broader accessibility and perceived reliability. Higher digital communication skills and better HRQoL were significant predictors of acceptability for both methods. Additionally, phone consultations had higher acceptability among patients with lower education and more frequent users. Socio-demographic factors such as age and urban or rural residence did not significantly affect acceptability.

These findings align with earlier studies showing telehealth modalities are generally well-received when matched with users’ capabilities and expectations [4,16,23,24]. Expanding upon previous work, our results emphasize the importance of evaluating acceptability across multiple domains, revealing distinct patterns in how different patient groups perceive each modality.

Patient portals were rated significantly higher in domains such as intervention coherence, support for smooth communication, and self-efficacy, particularly among individuals with greater digital communication skills and higher HRQoL. These results are consistent with previous studies which emphasize that users with strong digital literacy are more likely to perceive digital tools as intuitive, coherent, and effective in facilitating health communication [25,26]. The higher self-efficacy scores reflect confidence in using portals without encountering problems, highlighting the importance of user confidence as a determinant of continued engagement with patient portals [27,28].

In contrast, phone consultations were found to outperform patient portals in areas such as accessibility, reliability and perceived burden, particularly in terms of technical demands. This is consistent with evidence that phone-based modalities are more inclusive, particularly for individuals with limited digital literacy or cognitive capacity [29]. While video consultations may offer a richer interaction when feasible, telephone consultations remain a widely accessible and pragmatic alternative for routine follow-ups [30].

In terms of perceived burden, participants found phone consultations to be less time-consuming and easier to adopt, highlighting their value as low-complexity tools for reducing barriers to care, especially among older adults and individuals with limited formal education [29,30].

The role of functional health status also emerged as significant. In our multivariable models, HRQoL was a strong predictor of the acceptability of both modalities, particularly in the affective attitude domain. Poorer HRQoL often coincides with barriers such as low digital literacy, cognitive impairment, and high symptom burden, limiting engagement with technology-based care [31]. In our sample, lower HRQoL was more prevalent among individuals with lower education, skeletal disease, depression, anxiety, heart failure, pain/discomfort, or self-care difficulties. Such patterns are consistent with previous evidence identifying functional health status as a key driver of digital exclusion [32]. Among patients with activity limitations, reduced acceptability may stem from cognitive overload or anxiety associated with managing care remotely [33,34].

Digital communication skills, a core element of digital literacy, were also a strong predictor of acceptability for both portals and phone consultations. These skills directly influence patients’ confidence and comfort in engaging with digital health tools. Higher literacy reduces usability barriers and increases perceived ease of use [35,36], whereas low literacy can discourage portal use even among motivated patients [37,38]. As only communication skills were assessed, the influence of other digital abilities, such as information management, problem-solving, or online safety remains unknown and warrants further study.

Lower education predicted higher acceptability of telephone consultations but not patient portals. This contrasts with previous studies linking lower education to lower digital health engagement [39,40,41]. In our sample, patients with lower education rated several domains of acceptability significantly higher for both tools, with the largest difference in intervention coherence, reflecting their understanding of the intervention’s purpose. A possible explanation is that individuals with less formal education may have reduced expectations of digital technologies and may find phone consultations more familiar, intuitive, and aligned with their communication preferences [41,42].

Finally, a higher frequency of annual phone consultations was associated with greater acceptability of this modality. This finding suggests that familiarity and repeated exposure enhance patient comfort and perceived utility of remote care [4,43,44,45,46]. Regular use may contribute to normalization, leading patients to view phone consultations as an integral component of routine care rather than a substitute for in-person interaction [47]. For instance, one study reported that patients with more than 12 consultations per year were substantially more likely to accept phone calls for psychological support or administrative purposes such as work or sickness certifications [48].

### 4.2. Implications for Practice

The variability in acceptability across patient characteristics such as HRQoL, digital communication skills, education, and usage frequency indicates that one-size-fits-all digital health solutions are insufficient. Healthcare providers and system designers should tailor digital solutions to match patients’ individual capabilities and preferences. Patients with limited digital communication skills or functional impairments may benefit more from simpler options, such as phone consultations, rather than relying solely on patient portals. In contrast, those with stronger digital skills may prefer hybrid pathways that combine portals and phone consultations in place of face-to-face visits when an in-person examination is unnecessary.

In many countries, video consultations have emerged as an additional option, offering richer interaction than phone consultations while reducing the need for in-person visits. Although not currently widely implemented in Slovenia, incorporating such modalities in the future could further expand patient choice and improve flexibility in care delivery [30]. However, bypassing simpler tools in favor of advanced digital tools alone may be detrimental, as it risks widening disparities and exacerbating digital exclusion.

Given the strong association between digital communication skills and acceptability, health systems should actively invest in programs that enhance these skills, particularly among older adults, individuals with lower education levels, and socially disadvantaged groups. Such initiatives could include training workshops, step-by-step guidance on portal use, technical help lines, and digital navigation services embedded within primary care settings [49].

Equally important is supporting healthcare providers by offering training and resources to integrate digital tools effectively into their practice and tailor care based on patient preferences. This support helps reduce provider burden and ensures better patient engagement with digital services [50].

In the context of fragmented digital health in Slovenian primary care, developing a centralized and interoperable eHealth system could enhance patient-provider interactions and improve the acceptability of patient portals and phone consultations, while also providing a potential foundation for the safe and regulated introduction of video consultations in the future.

### 4.3. Strengths and Limitations

Strengths of this study include its multicenter design across Slovenian family medicine settings and its focus on a region with mature portal adoption. Furthermore, the use of the TFA allowed valid assessment of acceptability beyond proxies like satisfaction or frequency of use.

Limitations of this study involve the cross-sectional design, precluding causal inference, and convenience sampling, which may exclude non-users and introduce selection bias. Rather than applying multiple imputations, questionnaires with missing primary outcome data were excluded to maintain data integrity. However, comparison between excluded and included patients showed no significant differences in socio-demographic or clinical characteristics. Secondly, although the study involved multiple centers, the findings cannot be fully generalized to the Slovenian population given the uneven implementation of digital tools across clinical settings. With the development of a centralized eHealth system, patient experiences are expected to become more consistent and standardized. Thirdly, while acceptability was comprehensively assessed using the TFA, the study did not apply a dedicated Quality of Experience (QoE) framework to capture digital-specific factors such as interface design, system responsiveness, and interaction quality. Future research should combine TFA-based assessments with QoE measures and qualitative approaches to provide a more nuanced understanding of how design and performance influence user adoption and sustained engagement. Lastly, only digital communication skills were assessed rather than the broader concept of digital literacy, capturing only part of the competencies needed for full digital engagement.

## 5. Conclusions

Patient portals and phone consultations were generally acceptable to patients, though individual preferences varied based on key personal characteristics. Lower acceptability was associated with poorer HRQoL and limited digital communication skills, indicating significant barriers for more vulnerable groups. In contrast, patients with more frequent use of phone consultations and lower educational attainment reported higher acceptability, suggesting that familiarity and simplicity enhance comfort with this mode of care. These findings highlight the need to design and implement tailored digital health solutions that address patients’ diverse abilities and preferences. To promote equitable access and engagement, future research should focus on improving digital literacy, developing interventions to reduce digital exclusion, and assessing the effectiveness of hybrid care models across varied patient populations. Additionally, further studies on digital tool usage patterns and reasons for use or non-use could deepen understanding of barriers and facilitators among different groups.

## Figures and Tables

**Figure 1 healthcare-13-02662-f001:**
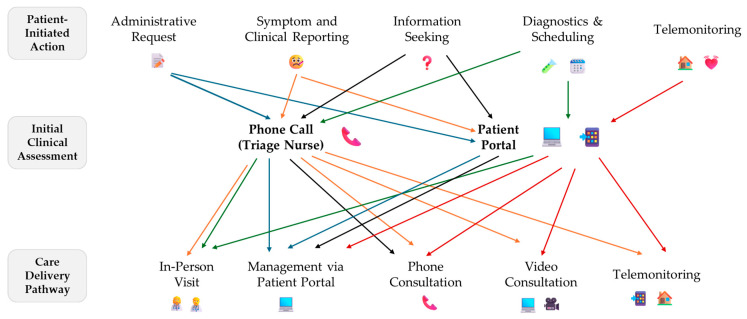
Hybrid primary care workflows in Slovenia showing how patient actions lead to different clinical assessments and care delivery modes. Video consultations and telemonitoring are still in the piloting phase.

**Figure 2 healthcare-13-02662-f002:**
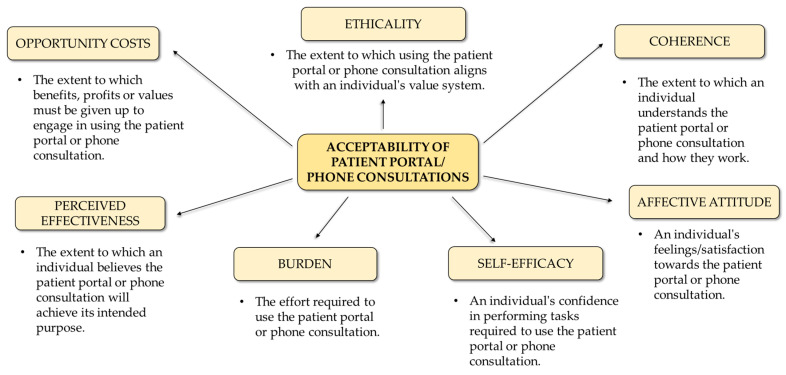
Theoretical framework of acceptability for the use of patient portals and phone consultations.

**Figure 3 healthcare-13-02662-f003:**
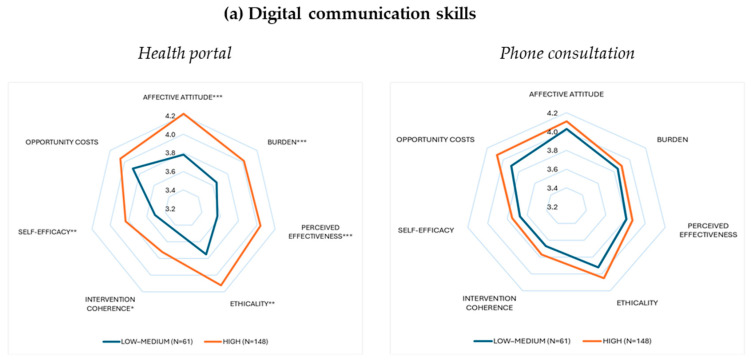
Radar plots showing mean scores across seven acceptability domains of patient portals and phone consultations, stratified by three important participant characteristics: (**a**) digital communication skills, (**b**) HRQoL, and (**c**) attained education level. Each subplot displays two groups with their respective sample sizes (N) in parentheses. Statistically significant group differences are indicated by asterisks as * *p* < 0.05, ** *p* < 0.01, and *** *p* < 0.001.

**Figure 4 healthcare-13-02662-f004:**
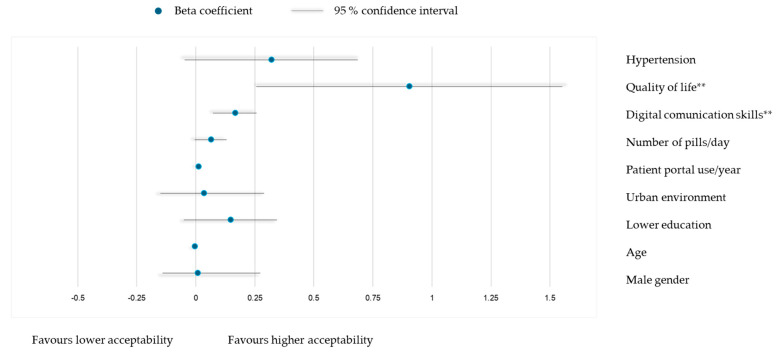
Multivariable linear model for association of TFA scores for using patient portals with socio-demographic and clinical characteristics: adjusted R^2^ = 0.182, model *p* < 0.001. ** *p* < 0.01.

**Figure 5 healthcare-13-02662-f005:**
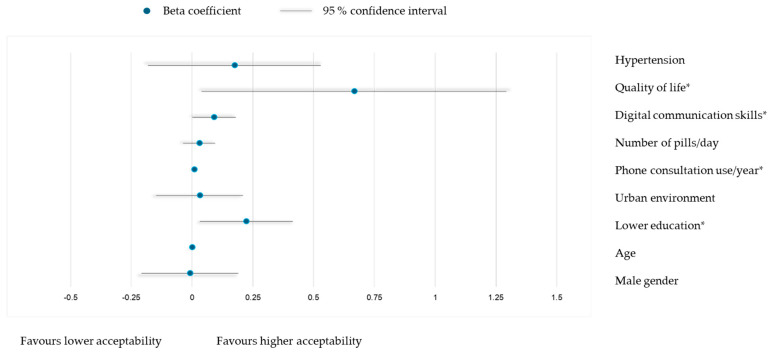
Multivariable linear model for association of TFA scores for using phone consultations with socio-demographic and clinical characteristics: adjusted R^2^ = 0.102, model *p* = 0.003. * *p* < 0.05.

**Table 1 healthcare-13-02662-t001:** Comparison in socio-demographic and clinical characteristics among those with high and low acceptability of using patient portals and phone consultations.

		Patient Portals			Phone Consultations		
Variable		Low Acceptability (n = 58)	Moderate–High Acceptability (n = 156)	*p*	Low Acceptability (n = 61)	Moderate–High Acceptability (n = 153)	*p*
**Age, n (%)**							
18–24 years	19 (8.9)	4 (6.9)	15 (9.6)	0.805	4 (6.6)	15 (9.8)	0.143
25–34 years	49 (22.9)	14 (24.1)	35 (22.4)		19 (31.1)	30 (19.6)	
35–44 years	66 (30.8)	17 (29.3)	49 (31.4)		15 (24.6)	51 (33.3)	
45–54 years	36 (16.8)	12 (20.7)	24 (15.4)		11 (18.0)	25 (16.3)	
55–64 years	22 (10.3)	7 (12.1)	15 (9.6)		9 (14.8)	13 (8.5)	
≥65 years	22 (10.3)	4 (6.9)	18 (11.5)		3 (4.9)	19 (12.4)	
**Gender, n (%)**							
Male	83 (38.8)	16 (27.6)	67 (42.9)	0.040	16 (26.2)	67 (43.8)	0.017
Female	131 (61.2)	42 (72.4)	89 (57.1)		45 (73.8)	86 (56.2)	
**Education level, n (%)**							
Primary school	20 (9.3)	3 (5.2)	17 (10.9)	0.436	1 (1.6)	19 (12.4)	0.018
Secondary school	111 (51.9)	28 (48.3)	83 (53.2)		28 (45.9)	83 (54.2)	
College	35 (16.4)	13 (22.4)	22 (14.1)		12 (19.7)	23 (15.0)	
University	32 (15.0)	10 (17.2)	22 (14.1)		15 (24.6)	17 (11.1)	
Master’s degree	16 (7.5)	4 (6.9)	12 (7.7)		5 (8.2)	11 (7.2)	
**Living area, n (%)**							
Urban	104 (48.6)	26 (44.8)	78 (50.0)	0.501	31 (50.8)	73 (47.7)	0.681
Rural	110 (51.4)	32 (55.2)	78 (50.0)		30 (49.2)	80 (52.3)	
**Digital communication skills,** **median (IQR)**	5 (3–5)	4 (3–5)	4 (3–5)	0.027	5 (4–5)	5 (4–5)	0.894
**Annual patient portal use,** **median (IQR)**	2 (1–4)	1 (1–4)	4 (1–12)	0.001	2 (1–12)	4 (1–4)	0.128
**Annual phone consultations use, median (IQR)**	1 (1–4)	1 (1–2)	1 (1–4)	0.385	1 (1–4)	1 (1–2)	0.004
**Hypertension,** **n (%)**	19 (8.9)	1 (1.7)	18 (11.5)	0.023	1 (1.6)	18 (11.8)	0.017
**Diabetes, n (%)**	14 (6.5)	3 (5.2)	11 (7.1)	0.599	2 (3.3)	12 (7.8)	0.212
**Depression, n (%)**	12 (5.6)	2 (3.4)	10 (6.4)	0.387	4 (6.6)	8 (5.2)	0.728
**Anxiety, n (%)**	9 (4.2)	4 (6.9)	5 (3.2)	0.244	3 (4.9)	6 (3.9)	0.765
**Number of pills per day, median (IQR)**	0 (0–2)	0 (0–1)	1 (0–3)	0.002	0 (0–1)	1 (0–3)	0.017
**EQ-5D utility score, median (IQR)**	0.96 (0.85–1.00)	0.94 (0.85–1.00)	1.00 (0.90–1.00)	0.058	0.96 (0.85–1.00)	1.00 (0.90–1.00)	0.044
**EQ VAS score, median (IQR)**	80 (60–90)	80 (60–90)	80 (60–90)	0.442	70 (50–90)	85 (70–95)	0.023

n, number; IQR, interquartile range; EQ-5D, EuroQol five-dimension scale; EQ VAS, EuroQol visual analogue scale.

**Table 2 healthcare-13-02662-t002:** Comparison of acceptability dimension scores between patient portals and phone consultations.

Dimension	Patient Portals (n = 214), Mean (95% CI)	Phone Consultations (n = 214), Mean (95% CI)	*p*
**1 Affective attitude**			
*1.1 Satisfaction with ability to contact clinic* via *portal or phone*	4.1 (4.0–4.3)	4.2 (4.1–4.3)	0.443
*1.2 Satisfaction with service process* via *portal or phone*	4.1 (4.0–4.2)	4.0 (3.9–4.1)	0.519
*1.3 Satisfaction with quality of service* via *portal or phone*	4.0 (3.9–4.2)	4.0 (3.9–4.1)	0.923
**2 Burden**			
*2.1 Simplicity of using the portal or phone*	4.0 (3.9–4.1)	4.1 (3.9–4.2)	0.673
*2.2 Minimal time required to use the portal or phone*	4.0 (3.9–4.1)	3.8 (3.7–4.0)	0.004
*2.3 Minimal effort needed to learn to use the portal or phone*	3.7 (3.5–3.8)	3.8 (3.6–3.9)	0.139
**3 Perceived effectiveness**			
*3.1 Effectiveness of the portal or phone in resolving health issues*	3.9 (3.8–4.0)	3.9 (3.7–4.0)	0.636
*3.2 Ability of the portal or phone to shorten clinic response time*	3.9 (3.8–4.1)	3.9 (3.7–4.0)	0.173
**4 Ethicality**			
*4.1 Safety and respect for privacy when using the portal or phone*	4.0 (3.9–4.1)	4.0 (3.9–4.1)	0.900
**5 Coherence**			
*5.1 Support for smooth communication* via *the portal or phone*	3.8 (3.7–3.9)	3.7 (3.6–3.8)	0.028
*5.2 Accessibility of the portal or phone regardless of digital skills*	3.5 (3.3–3.6)	3.8 (3.7–3.9)	<0.001
**6 Self-efficacy**			
*6.1 Confidence in communicating* via *the portal or phone*	4.0 (3.9–4.1)	3.9 (3.8–4.0)	0.002
*6.2 Confidence in using the portal or phone without problems*	3.5 (3.3–3.6)	3.6 (3.5–3.7)	<0.001
**7 Opportunity costs**			
*7.1 Time saved for other activities through portal or phone use*	4.0 (3.9–4.1)	4.0 (3.9–4.1)	0.782
*7.2 Reduction in logistical barriers through portal or phone use*	4.1 (4.0–4.1)	4.1 (4.0–4.2)	0.258
**Mean combined scores (95% CI)**	**3.9** **(3.8–4.0)**	**3.9** **(3.8–4.0)**	**0.923**

## Data Availability

The data underlying the study results are freely available from the Harvard Dataverse repository at: https://doi.org/10.7910/DVN/EUDLZW (accessed on 11 August 2025).

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
