# Peer review of "Acceptability of Patient Portals and Phone Consultations in Hybrid Primary Care: A Slovenian Multi-Centre Pilot Study"

_healthcare, 2025, doi:10.3390/healthcare13212662_

Round 1

Reviewer 1 Report

Comments and Suggestions for Authors

The study an assessement which assess factors and barrieres to the Digital communication.

Althrough, the merit of study, the assessment of  the aceptability should includ assessing the QoE. A Bad QoE may drive the user to reject the Tools.

I therefore recommend the author to revise their work including my comments or to argure why they did not assess th QoE.

Though, to assess the acceptability of an online it solution, it us important to assess the user experience using the solution 1. What is the network Kalender 2. What is the Level of the quality of service provided by the healthcare professionals  3. Etc.   In the case, a Prior study alread did this assessment, the paper has to refer to this and consider it in the article.  

Author Response

The study an assessement which assess factors and barrieres to the Digital communication. Althrough, the merit of study, the assessment of  the aceptability should includ assessing the QoE. A Bad QoE may drive the user to reject the Tools. I therefore recommend the author to revise their work including my comments or to argure why they did not assess th QoE.  Though, to assess the acceptability of an online it solution, it us important to assess the user experience using the solution 1. What is the network Kalender 2. What is the Level of the quality of service provided by the healthcare professionals  3. Etc.   In the case, a Prior study alread did this assessment, the paper has to refer to this and consider it in the article.  

ANSWER: Thank you for your thoughtful comment. Quality of Experience (QoE) is indeed an important dimension in the evaluation of digital communication tools, as it captures how users perceive usability, accessibility, comfort, and trust in addition to technical performance. In our study, we applied the Theoretical Framework of Acceptability (TFA), which already includes several constructs that overlap with QoE, such as affective attitude, burden, perceived effectiveness, and self-efficacy. These domains reflect users’ subjective perceptions and therefore intersect conceptually with QoE.

We acknowledge, however, that a dedicated QoE assessment could provide additional nuance. TFA is a broader framework that also incorporates constructs such as ethicality, coherence, and opportunity costs, thereby extending beyond QoE in addressing contextual and value-based determinants of acceptability. In this sense, QoE can be regarded as a subset of acceptability, focusing specifically on the quality of the digital interaction itself, whereas TFA encompasses the wider range of factors influencing whether an intervention is judged acceptable overall.

Many QoE-related aspects—such as usability, satisfaction, perceived effort, and trust/privacy—are already represented within the TFA constructs used in our questionnaire. What is less explicitly addressed are digital-specific QoE elements such as interface design (e.g., navigation ease, error recovery) and technical smoothness (e.g., connection stability, speed, audiovisual clarity). We have now clarified this point in the limitations section, noting that while TFA captures much of QoE, a dedicated QoE framework might further enrich future evaluations.

Reviewer 2 Report

Comments and Suggestions for Authors

Abstract
The abstract is clear and easy to understand the research summary.

Introduction
The introduction is written clearly and provides relevant literature to support the research narrative.

Materials and methods
The research approach is adequately described. However, it is recommended that the training elements be explained, particularly how you ensure respondents can use the application smoothly. For instance, consider providing just-in-time online help via a chatbot or short video.

Result
The results are described comprehensively. The tables and figures provide relevant support to research questions.

Discussion
You mentioned that the health information system in Slovenia has been implemented in a fragmented manner. In other words, there are significant issues with the centralization and integration of various electronic medical record (EMR) applications, which impacts  the implementation of telemedicine in Slovenian health centres. Although the primary focus of the research is to study the acceptability of patient portals and phone consultations, it is also essential to discuss and include the central EMR to ensure seamless patient consultations.

Conclusion
Adequate explanation.

Author Response

Abstract
The abstract is clear and easy to understand the research summary.

ANSWER: Thank you for your comment.

Introduction
The introduction is written clearly and provides relevant literature to support the research narrative.

ANSWER: Thank you for your comment.

Materials and methods
The research approach is adequately described. However, it is recommended that the training elements be explained, particularly how you ensure respondents can use the application smoothly. For instance, consider providing just-in-time online help via a chatbot or short video.

ANSWER: Thank you for your comment. We have added a detailed description of the training elements in the “Study Setting” section, specifying how healthcare providers were trained by patient portal providers and subsequently offered one-on-one guidance to patients. Additionally, we clarified the availability of supportive resources, including community workshops and online help, to ensure that respondents could use the application smoothly.

Result
The results are described comprehensively. The tables and figures provide relevant support to research questions.

ANSWER: Thank you for your comment.

Discussion
You mentioned that the health information system in Slovenia has been implemented in a fragmented manner. In other words, there are significant issues with the centralization and integration of various electronic medical record (EMR) applications, which impacts  the implementation of telemedicine in Slovenian health centres. Although the primary focus of the research is to study the acceptability of patient portals and phone consultations, it is also essential to discuss and include the central EMR to ensure seamless patient consultations.

ANSWER: Thank you for your comment. We agree that the fragmented implementation of health information systems in Slovenia, including limited centralization and integration of EMR applications, can impact telemedicine delivery. While our study primarily focused on the acceptability of patient portals and phone consultations, we have now included a discussion in the revised manuscript emphasizing that developing a centralized and interoperable eHealth system could enhance patient-provider interactions and improve the acceptability and seamless use of digital communication tools in routine clinical practice.

Conclusion
Adequate explanation.

ANSWER: Thank you for your comment.

Reviewer 3 Report

Comments and Suggestions for Authors

This article assesses the acceptance of patient portals and telephone consultations in mixed primary care. An instrument developed based on the Acceptance Theoretical Framework (TFA) was used to measure acceptance. With the rapid development of digital technology, this manuscript has good theoretical and practical significance. But for now, there are still many problems to be improved.
1 This manuscript can be considered as a study related to the use of digital technology, and factors such as the patient's digital literacy are taken into account. That's very good. But a question worth exploring is whether telephone counseling should be included. Or what is the rationale for doing so? If from the perspective of a hybrid patient portal, it seems to be acceptable. But in terms of selected factors (digital literacy, digital communication), does the inclusion of telephone consultation introduce bias? Because these factors seem to have little to do with telephone counseling. I think the author should explain the reasons for including telephone consultation in more detail, especially in relation to digital literacy and digital communication skills.

2 Inclusion and exclusion criteria should be more clearly described.

3 The process of data collection should also be described in detail. For now, the collection process is very vague.

4 It seems not very reasonable to use 214 respondents to estimate the overall sample. More evidence is needed to justify such estimates. Such as sampling methods, consistency of sample characteristics, etc.

5 The statistical analysis was acceptable.

6 Discussion of the findings is inadequate.
(1) The results of the study need to be more fully analyzed.
(2) More sufficient evidence and inferences should be added when comparing the differences with previous studies.

Author Response

This article assesses the acceptance of patient portals and telephone consultations in mixed primary care. An instrument developed based on the Acceptance Theoretical Framework (TFA) was used to measure acceptance. With the rapid development of digital technology, this manuscript has good theoretical and practical significance. But for now, there are still many problems to be improved.

1 This manuscript can be considered as a study related to the use of digital technology, and factors such as the patient's digital literacy are taken into account. That's very good. But a question worth exploring is whether telephone counseling should be included. Or what is the rationale for doing so? If from the perspective of a hybrid patient portal, it seems to be acceptable. But in terms of selected factors (digital literacy, digital communication), does the inclusion of telephone consultation introduce bias? Because these factors seem to have little to do with telephone counseling. I think the author should explain the reasons for including telephone consultation in more detail, especially in relation to digital literacy and digital communication skills.

ANSWER: Thank you for your comment. We included telephone consultations alongside patient portals because our aim was to assess the acceptability of digital and remote communication modalities broadly in primary care, reflecting the hybrid nature of care delivery where both online and phone-based interactions are commonly used. While telephone consultations do not require digital literacy per se, they still involve aspects of communication skills and user confidence in interacting with healthcare providers remotely. Including telephone consultations allows us to compare user acceptability across different modalities and to understand how digital literacy and communication skills may differentially influence the use of digital versus non-digital remote services. We have clarified this rationale in the revised manuscript to explain the relevance of telephone consultations in the context of our study.

2 Inclusion and exclusion criteria should be more clearly described.

ANSWER: Thank you for your comment. We have revised the Methods section to provide a clearer description of the inclusion and exclusion criteria. The revised text now specifies the requirements for prior use of both patient portals and phone consultations, access to necessary technology, language proficiency, and the conditions under which patients were excluded.

3 The process of data collection should also be described in detail. For now, the collection process is very vague.

ANSWER: Thank you for your comment. We have expanded the description of the data collection process in the Methods section, detailing the recruitment procedure, setting, and timeline, as well as how surveys were administered and completed. This provides greater transparency and allows readers to better assess the rigor of the study design.

4 It seems not very reasonable to use 214 respondents to estimate the overall sample. More evidence is needed to justify such estimates. Such as sampling methods, consistency of sample characteristics, etc.

ANSWER: Thank you very much for your comment. We have clarified the sampling approach in the Methods section. The study was designed with a predefined target of 250 participants, recruited consecutively using convenience sampling. The required sample size was calculated as 245, based on previous studies reporting low variability in TFA scores, a 95% confidence level, and a margin of error of ±0.05. The relatively low variability in scores contributed to a smaller required sample size.

In total, 259 patients were recruited; however, 45 surveys were excluded from analysis due to incomplete primary outcome data (TFA). To ensure transparency, we compared the socio-demographic and clinical characteristics of included and excluded patients and found no significant differences, supporting the consistency and representativeness of the analyzed sample. We acknowledge, however, that the final sample of 214 participants may somewhat limit generalizability, which is consistent with the pilot nature of the study and has been explicitly addressed in both the Methods and Limitations sections.

5 The statistical analysis was acceptable.

ANSWER: Thank you for your comment.

6 Discussion of the findings is inadequate.
(1) The results of the study need to be more fully analyzed.
(2) More sufficient evidence and inferences should be added when comparing the differences with previous studies.

ANSWER: We have substantially revised and expanded the discussion section to provide a deeper analysis of our findings. In particular, we have included additional evidence and clearer inferences when comparing our results with previous studies.

Reviewer 4 Report

Comments and Suggestions for Authors

The paper presents an interesting discussion about the acceptability of digital health tools in Slovenia: the patient portal and telephone consultations. The paper is well-written and consistent. Yet in the introduction, the authors state that video call consultations are not allowed in Slovenia. I suggest going into this more in the introduction and discussion. Despite technological advances and the imposition of limits, face-to-face consultations during the COVID-19 pandemic have shortened the discussion about incorporating digital technologies worldwide. In several countries, teleconsultations were briefly regulated, given the need to do so in the face of the COVID-19 pandemic. Why are video calls not allowed in medical consultations? 
The use of these digital services was related to educational level and personal preferences. In the study limitations, it´s essential to cite the research design: a qualitative approach would help to identify the barries to digital health deeply. 

Author Response

The paper presents an interesting discussion about the acceptability of digital health tools in Slovenia: the patient portal and telephone consultations. The paper is well-written and consistent. Yet in the introduction, the authors state that video call consultations are not allowed in Slovenia. I suggest going into this more in the introduction and discussion. Despite technological advances and the imposition of limits, face-to-face consultations during the COVID-19 pandemic have shortened the discussion about incorporating digital technologies worldwide. In several countries, teleconsultations were briefly regulated, given the need to do so in the face of the COVID-19 pandemic. Why are video calls not allowed in medical consultations? The use of these digital services was related to educational level and personal preferences.

ANSWER: Thank you for your valuable comment. In this study, we focused on evaluating digital tools that have already been formally implemented and integrated into clinical workflows, namely the patient portal and telephone consultations. At present, video consultations are not part of the public healthcare system in Slovenia; they are primarily available as out-of-pocket services in the private sector. However, this situation is expected to change soon. A new national module for video consultations is currently under development and is planned for integration into the public healthcare system in the coming year. We clarified this point in the healthcare context section and discussion (section Implications for practice) to better explain the regulatory context and the reasons why video consultations have not yet been widely adopted in Slovenia.

In the study limitations, it´s essential to cite the research design: a qualitative approach would help to identify the barries to digital health deeply. 

ANSWER: Thank you for your comment. We agree that a qualitative approach could provide deeper insights into the barriers to digital health. We have now added this point to the limitations section, highlighting that future research using qualitative methods would be valuable to explore barriers and user experiences in greater depth.

Reviewer 5 Report

Comments and Suggestions for Authors

Dear Authors, thank you for your work. I appreciated all section, but -I think- you can improved methodological section with methodology flow chart. You should add a flow chart for patients flow, too. results are ok, with a good textual and iconographic description, and all other section are well rappresentated

Author Response

Dear Authors, thank you for your work. I appreciated all section, but -I think- you can improved methodological section with methodology flow chart. You should add a flow chart for patients flow, too. results are ok, with a good textual and iconographic description, and all other section are well rappresentated.

 ANSWER: Thank you for your thoughtful comment and valuable suggestions regarding the inclusion of methodological and patient flow charts. Your feedback motivated us to restructure the description of the healthcare context and the hybrid primary care model, including detailed primary care workflows in Slovenia. We believe these additions, together with the improved methodological description, will help readers better understand the complexity of the evaluated digital tools.

Round 2

Reviewer 1 Report

Comments and Suggestions for Authors

The authors worked out the review comments and clarify my concern regarding the QoE assessment.

Reviewer 3 Report

Comments and Suggestions for Authors

经过这次修订,此手稿可以考虑接受。

Reviewer 5 Report

Comments and Suggestions for Authors

your work is appreciated. thank you